# Oxygen vacancy associated single-electron transfer for photofixation of $CO_2$ to long-chain chemicals

Shichuan Chen[1], Hui Wang[1], Zhixiong Kang[1], Sen Jin[1], Xiaodong Zhang [1], Xusheng Zheng[2], Zeming Qi[2], Junfa Zhu [2], Bicai Pan[1] & Yi Xie[1]

The photofixation and utilization of $CO_2$ via single-electron mechanism is considered to be a clean and green way to produce high-value-added commodity chemicals with long carbon chains. However, this topic has not been fully explored for the highly negative reduction potential in the formation of reactive carbonate radical. Herein, by taking $Bi_2O_3$ nanosheets as a model system, we illustrate that oxygen vacancies confined in atomic layers can lower the adsorption energy of $CO_2$ on the reactive sites, and thus activate $CO_2$ by single-electron transfer in mild conditions. As demonstrated, $Bi_2O_3$ nanosheets with rich oxygen vacancies show enhanced generation of $\cdot CO_2^-$ species during the reaction process and achieve a high conversion yield of dimethyl carbonate (DMC) with nearly 100% selectivity in the presence of methanol. This study establishes a practical way for the photofixation of $CO_2$ to long-chain chemicals via defect engineering.

[1] Hefei National Laboratory for Physical Sciences at the Microscale, Collaborative Innovation Center of Chemistry for Energy Materials, University of Science and Technology of China, Hefei 230026, P.R. China. [2] National Synchrotron Radiation Laboratory, University of Science and Technology of China, Hefei 230029, P.R. China. These authors contributed equally: Shichuan Chen, Hui Wang. Correspondence and requests for materials should be addressed to X.Z. (email: zhxid@ustc.edu.cn) Y.X. (email: yxie@ustc.edu.cn)

Continuous accumulation of carbon dioxide ($CO_2$) in the atmosphere represents a major contributor to climate change through global warming[1,2]. In the past decades, the fixation and utilization of $CO_2$ has attracted great research interests all over the world, where the waste $CO_2$ could be converted to valuable chemicals for further use[3–5]. In view of the strong carbon–oxygen bonds among the no electric dipole molecules, the conversion of inert $CO_2$ to other chemicals has been often performed under high temperature and pressure[6–8]. Compared to traditional thermal based processes, light-driven $CO_2$ fixation may be a clean and cost-effective strategy to solve the energy issues and environmental problems[9–13].

Because of the highly negative reduction potential for the formation of reactive carbonate radical ($\bullet CO_2^-$, $-1.9$ V vs NHE), the photoreduction of $CO_2$ often undergoes a proton-assisted multi-electrons transfer process to overcome the high energies[14,15]. However, the process is unaccessible for producing high-value-added organic chemicals with chains longer than three carbons, which could possess wide industrial applications and great economic value. For example, one of the most important long-chain chemicals, dimethyl carbonate (DMC), is a versatile and green chemical regent, and has been widely used as fuel additive, electrolyte in lithium-ion batteries, monomer for organic synthesis, etc[16–18]. Up to now, the synthetic routes of DMC or other long-chain chemicals often undergo high temperature and use of toxic and explosive agents, such as phosgene, hydrogen chloride, and carbon monoxide[17,18]. Compared to traditional strategies, the photofixation of $CO_2$ to long-chain chemicals would be extremely attractive for its green and environmental friendly nature. In that case, it is essential for us to design the light harvesting semiconductors capable of activating $CO_2$ molecules with a single electron to the direct synthesis of long-chain chemicals, which is also the rate-determining step for $CO_2$ photofixation[19,20].

Bearing it in mind, we pay our attention to the catalysts with surface deficiency, where the introduced defects serve as reactive centers to adsorb gas molecules and hence activate them by lowering the adsorption energies. To go further, for the presence of abundant localized electrons in the defect sites, the enhanced charge transfer between catalysts and adsorbates could be realized. Thus, we propose that the atomically thin layers with rich surface defects and fully exposed active sites shall be an ideal structural model of pursuing high $CO_2$ photofixation efficiency.

## Results

**Theoretical studies**. Herein, by taking the $Bi_2O_3$ atomic layers with oxygen vacancies (OVs) as an example, we studied the role of surface deficiency in $CO_2$ photoactivation, whose electronic structures and adsorption with $CO_2$ were investigated via density functional theory (DFT) calculations. As seen from the calculated density of states (DOS), the presence of OVs results in the appearance of a new defect level in $Bi_2O_3$ atomic layers, which is beneficial to the photoexcitation of electrons to the conduction band (Supplementary Fig. 1)[21,22]. In addition, the differences of charge density between the $Bi_2O_3$ atomic layers and their defective structure (lower part of Fig. 1) clearly indicate that the electrons neighboring OVs could be localized, suggesting the electrons in the defect structures are more likely to be excited[19]. To go further, the $CO_2$ adsorption on the $Bi_2O_3$ atomic layers with and without OVs was performed to gain insights into the role of OVs in the chemisorption processes. As expected, the $Bi_2O_3$ atomic layers with OVs show promising $CO_2$ adsorption ability with negative adsorption energy of about $-0.30$ eV, while the chemisorption of $CO_2$ on the perfect $Bi_2O_3$ atomic layers was ruled out because of the weak interaction (Fig. 1). As shown in

the lower part of Fig. 1, the yellow and blue isosurfaces represent charge accumulation and depletion in the space, respectively, indicating the presence of exchange and transfer of electrons between the OVs and $CO_2$. Therefore, based on the above advantages, the $Bi_2O_3$ atomic layers with rich OVs would be efficient catalysts to overcome the bottlenecks of carbonate radical generation for $CO_2$ photofixation.

**Synthesis and characterization**. In this study, $Bi_2O_3$ atomic layers were prepared via in situ oxidation of freshly exfoliated Bi nanosheets owing to their high surface energy[23], during which oxygen deficiencies could be readily introduced (Fig. 2a). Meanwhile, the short in situ oxidation time benefits the generation of a high concentration of OVs. By taking the sample synthesized with 1 h oxidation as an example, the X-ray diffraction (XRD) pattern of its collected powers could be readily indexed to $Bi_2O_3$ (JCPDS Card No. 71–0465) with high phase purity (Supplementary Fig. 2). The Fourier-transform infrared (FT-IR) spectra show that the intercalated amine in the Bi-amine hybrid intermediate was simultaneously removed during the exfoliation and in situ oxidation processes (Supplementary Fig. 3), suggesting the clean surface of as-obtained sample[24]. As shown in Fig. 2b of transmission electron microscopy (TEM) image, the synthetic sample shows a sheet-like morphology. In addition, atomic force microscopy (AFM) and corresponding height profiles of the nanosheets show an average thickness of 0.68 nm (Fig. 2c, d), which was consistent with the thickness of single-unit-cell $Bi_2O_3$ slab. The above results indicate that the $Bi_2O_3$ nanosheets with single-layer thickness were obtained.

To gain insights into the deficiencies in the as-prepared nanosheets, aberration-corrected high-angle annular dark-field scanning transmission electron microscopy (HAADF-STEM) was first carried out to show their fine structures. As displayed in Fig. 3a and Supplementary Fig. 4, the nanosheets show the interplanar spacings of 0.27 and 0.34 nm, corresponding to the distances of the (200) and (002) planes of $Bi_2O_3$, respectively, and indicating they are mostly enclosed by (010) faces. In addition, as compared to the sample with a long oxidation time of 6 h, the sample with short oxidation time shows obvious lattice disorder as labeled in Fig. 3a, which may be derived from the vacancies induced by the unsaturated coordination of metal atoms[25]. To further explore their defect structure, X-ray photoelectron spectroscopy (XPS) was employed to study the valence states of the samples, where the O $2p$ peak located at 529.8 eV is attributed to the lattice oxygen, while the other one located at 530.9 eV is the signal of oxygen atoms in the vicinity of an OV[25,26]. Notably, as clearly displayed in Fig. 3b, the sample with 1 h oxidation possesses more OVs than that with 6 h oxidation (Fig. 3b), which were labeled as oxygen vacancies-rich $Bi_2O_3$ nanosheet (OV-rich-$Bi_2O_3$) and oxygen vacancies-poor $Bi_2O_3$ nanosheet (OV-poor-$Bi_2O_3$), respectively. Furthermore, electron spin resonance (ESR) spectroscopy was performed to study the concentration of OVs in the samples. As shown in Fig. 3c, both the samples exhibited similar ESR signal ($g = 2.002$), which could be identified as the electrons trapped in OVs[25]. OV-rich-$Bi_2O_3$ shows a greatly enhanced intensity in ESR signal than OV-poor-$Bi_2O_3$. It is known that, OVs could emit photoluminescence (PL) under light excitation, thus PL spectra were further performed to investigate the defects in the samples. As seen in Fig. 3d, both samples show two distinct PL emissions that could be ascribed to the signal of recombination of photogenerated electron–hole pairs (443 nm) and OVs (613 nm)[26], respectively. The PL emission intensity of OV-rich-$Bi_2O_3$ is lower than OV-poor-$Bi_2O_3$, which implies that OVs could hinder the recombination of photogenerated carriers in the $Bi_2O_3$ nanosheets[5]. And OV-rich-$Bi_2O_3$ possesses higher

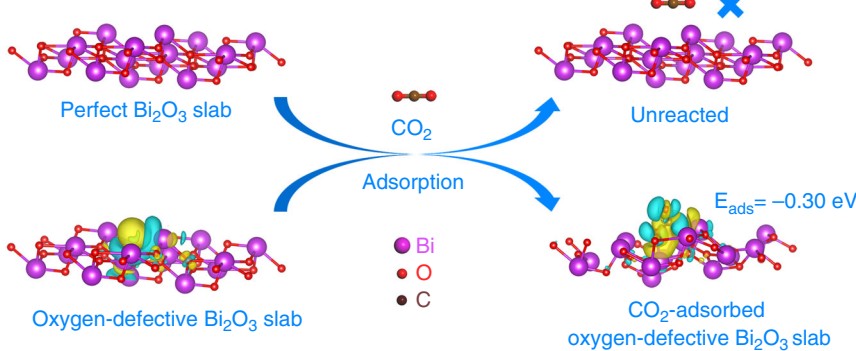

**Fig. 1** Theoretical study. Schematic illustration of the adsorption of $CO_2$ molecules onto perfect and oxygen-defective $Bi_2O_3$ single-unit-cell layer slab with the partial charge density of oxygen vacancies

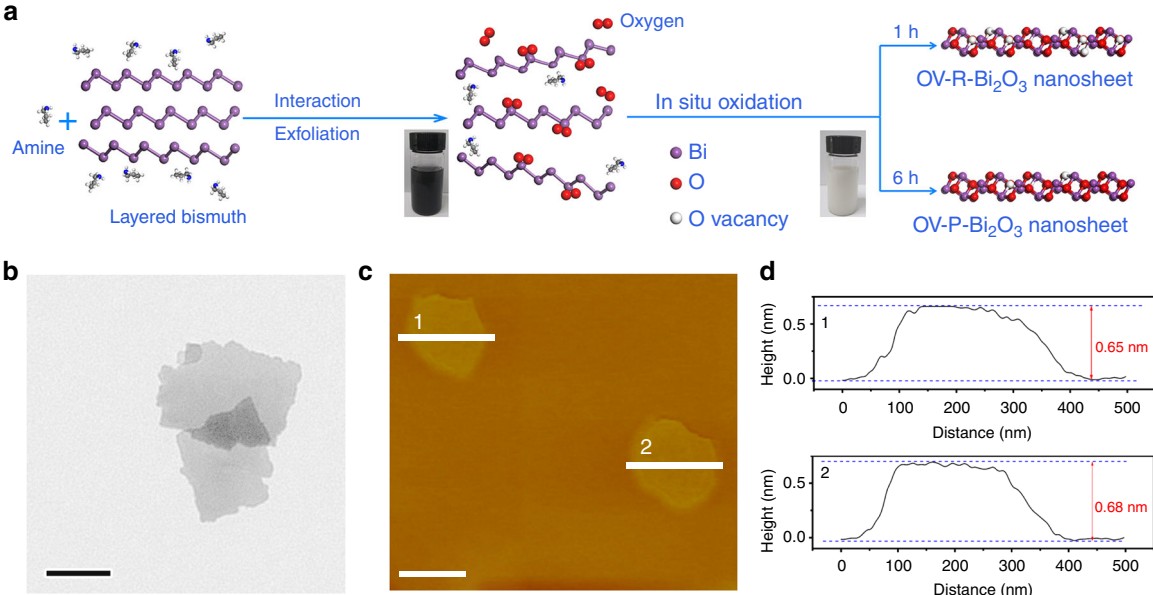

**Fig. 2** Synthesis and morphology study. **a** Schematic introduction for the preparation of ultrathin $Bi_2O_3$ nanosheets with rich/poor oxygen vacancies. **b** TEM image of the OV-rich-$Bi_2O_3$ nanosheets. **c**, **d** AFM image and the height distributions (close to the regions of Line 1 and 2) of the OV-rich-$Bi_2O_3$ nanosheets, respectively. The scale bars in (**b**, **c**) are 200 nm

PL intensity of the peak 613 nm compared to OV-poor-$Bi_2O_3$, which is in agreement with the XPS and ESR analysis. Hence, the collective results shown above prove that OV-rich-$Bi_2O_3$ nanosheets with an abundant amount of OVs have been successfully designed.

**Catalytic performances and mechanistic investigation**. As known, the presence of OVs could strongly affect the adsorption of gas molecules and the catalytic activity of the catalysts. Herein, by taking the photofixation of $CO_2$ to high-value-added DMC in the presence of $CH_3OH$ for an example, we systematically studied the catalytic ability of the $Bi_2O_3$ nanosheets with OVs. In situ diffuse reflectance infrared Fourier-transform spectroscopy (DRIFT) was carried out to investigate the local information for reaction species and their intermediates on the surface of the $Bi_2O_3$ nanosheets during the reaction[27–30]. As shown in Fig. 4a, after treating the OV-rich-$Bi_2O_3$ nanosheets in the mixture of $CO_2$ and $CH_3OH$ under light irradiation, three new peaks at 1294, 1456, and 1779 cm$^{-1}$ were generated, which could be assigned to the vibration of $\cdot CO_2^-$, carbonate-like ($CO_3^=$) species and DMC, respectively[27,29]. The intensities of the peaks gradually

increased with increasing reaction time. Meanwhile, as prolonging the reaction time to 60 min and the catalysis reaching a steady state, the reaction intermediate $\cdot CO_2^-$ is expected to level off while the final product DMC continues to rise (Fig. 4a). It is worth noting that, the formation of $\cdot CO_2^-$ species is the rate-determining step for the generation of DMC[31]. As expected, only OV-rich-$Bi_2O_3$ nanosheets could effectively activate $CO_2$ and initiate the reaction (Fig. 4b).

To further explore the intermediate states in the reaction, the adsorption ability of $CO_2$ in the surface of OV-rich-$Bi_2O_3$ was studied by quasi in situ XPS measurements under the simulated reaction conditions ($CO_2$ (0.2 MPa) at 373 K under Xe-lamp irradiation)[32,33]. As shown in Fig. 4c of the C 1s spectra, besides the intrinsic peak of carbon bonds at 284.8 eV, there was a new peak located at about 288.3 eV, corresponding to the signal of $\cdot CO_2^-$ species[34,35]. The experimental observations of $\cdot CO_2^-$ species indicates that the $Bi_2O_3$ nanosheets with OVs could be efficient catalysts for activating $CO_2$ via single-electron strategy. And it is anticipated that OV-rich-$Bi_2O_3$ show enhanced ability in the generation of $\cdot CO_2^-$ species compared to OV-poor-$Bi_2O_3$, corresponding well to the in situ DRIFT results. Based on the above observations, we can conclude that surface OVs in the

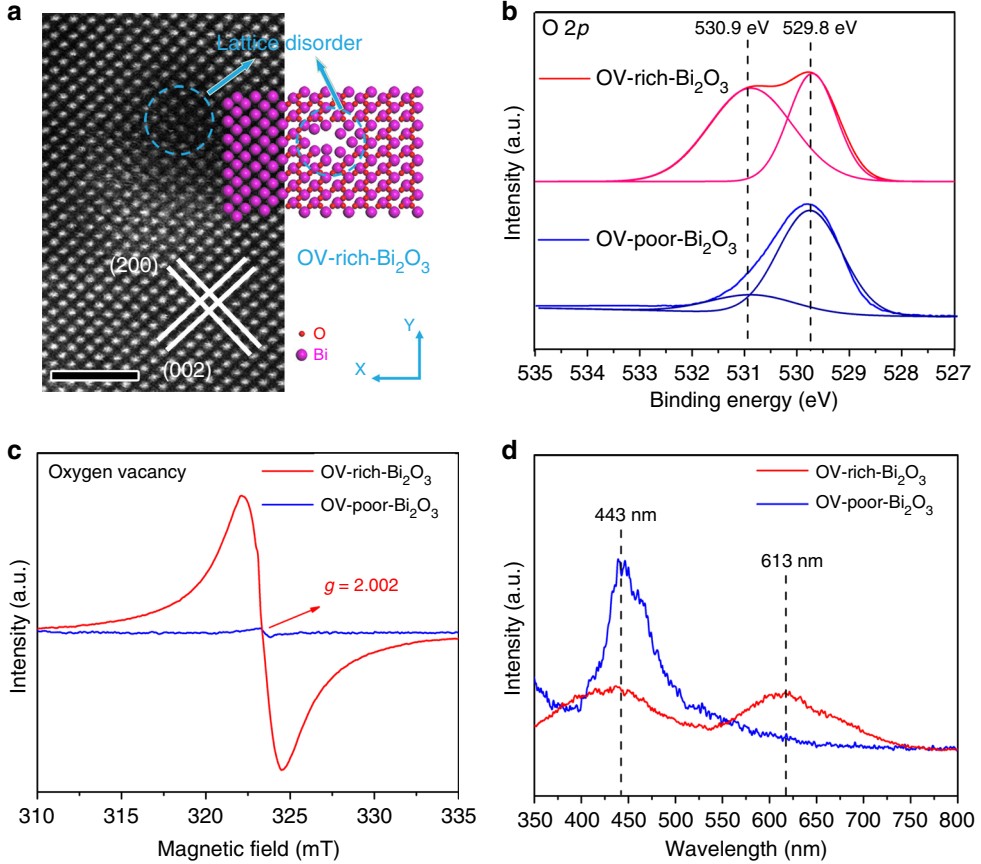

**Fig. 3** Structure characterizations for the defect-controlled $Bi_2O_3$ nanosheets. **a** Atomic-resolution HAADF-STEM images and corresponding structure model of OV-rich-$Bi_2O_3$ nanosheets. The scale bar is 2 nm. **b** O $2p$ XPS spectra, **c** Room-temperature ESR spectra, and **d** PL spectra of $Bi_2O_3$ nanosheets with rich and poor oxygen vacancies, respectively

$Bi_2O_3$ nanosheets could enhance the formation of $\cdot CO_2^-$ species and carbonate-like ($CO_3^=$) species, which are the most important intermediate states in the reactions of $CO_2$ conversion. To further understand the reaction mechanism, ESR spectroscopy was carried out to detect and identify the generated radical species during the reaction by using 5,5-dimethyl-1-pyrroline N-oxide (DMPO) as a trapping regent[36]. As displayed in Supplementary Fig. 5, the methanol solution with $Bi_2O_3$ nanosheets gave rise to a six-line spectrum, which can be attributed to the DMPO-$CH_3$ adducts, and thus it is reasonable to infer the generation of $\cdot CH_3$ intermediate in the reaction processes.

Based on the above analysis, the direct photogeneration of DMC from the reaction of $CO_2$ and $CH_3OH$ was conducted in acetonitrile ($CH_3CN$) solution with $CO_2$ pressure of 0.2 MPa at 373 K (the products were detected by nuclear magnetic resonance (NMR) spectroscopy, Supplementary Fig. 6). As shown in Fig. 5a and Supplementary Table 1, both OV-rich-$Bi_2O_3$ and OV-poor-$Bi_2O_3$ nanosheets could catalyze $CO_2$ and $CH_3OH$ to DMC with nearly 100% selectivity, despite trace amounts of DMC can be detected in the presence of bulk $Bi_2O_3$, being consistent with the result of Fig. 4b. Meanwhile, the conversion yield of OV-rich-$Bi_2O_3$ can reach to about 18 %, which is 9 times higher than that of OV-poor-$Bi_2O_3$, indicating the OVs in $Bi_2O_3$ nanosheets offer indispensable active sites for the reactions. The $^{13}CO_2$ labeling experiment is a useful tool to reveal the dominant product indeed originated from the photofixation of $CO_2$ or not. The isotope tracing experiments were performed using common $CO_2$ and $^{13}CO_2$ ($^{13}C$, 99 %), respectively. The $^{13}CO_2$ labelling experiments involving the same set-up and reaction condition both yielded a product that generates an obvious $^{13}C$ NMR peaks at 156.6-ppm and 54.8-ppm signals referring to DMC (Supplementary Fig. 6b)[37]. The labelling carbon at 156.6-ppm comes from carbon sources $CO_2$ and another labelling carbon at 54.8-ppm comes from $CH_3OH$. In the experiment, we employed the $^{13}CO_2$ as $^{13}C$-labeled carbon sources. The intensity of $^{13}CO_2$ that located at 156.6-ppm is much higher than that of common $CO_2$, thus clearly indicating that the product DMC is indeed derived from $CO_2$. The nearly identical BET surface area as displayed in Supplementary Fig. 7 further implied the crucial role of OVs in the catalytic performance of the $Bi_2O_3$ atomic layers. It is worth noting that, the high conversion yield and nearly 100% selectivity of OV-rich-$Bi_2O_3$ nanosheets under mild conditions are superior to most of previously reported catalysts, which were commonly conducted in high temperature and pressure (Supplementary Table 1). In order to further show the catalytic performances of our samples, we have compared OV-rich-$Bi_2O_3$ nanosheets with other well-established catalysts under the same reaction conditions, including $CeO_2$ nanosheets, $V_2O_5$ nanosheets and $ZrO_2$ nanoparticles[38–40]. As seen in the Supplementary Fig. 8, the conversion yield of OV-rich-$Bi_2O_3$ nanosheets was much higher than that of the other three catalysts, indicating the outstanding catalytic performances of OV-rich-$Bi_2O_3$ nanosheets for DMC generation. Moreover, OV-rich-$Bi_2O_3$ nanosheets exhibited high stability after 25 catalytic cycles up to 200 h in Fig. 5b, during which their morphology and defective structure were also retained (Supplementary Fig. 9). Thus, OV-rich-$Bi_2O_3$ nanosheets were efficient catalysts for photocatalytic DMC production.

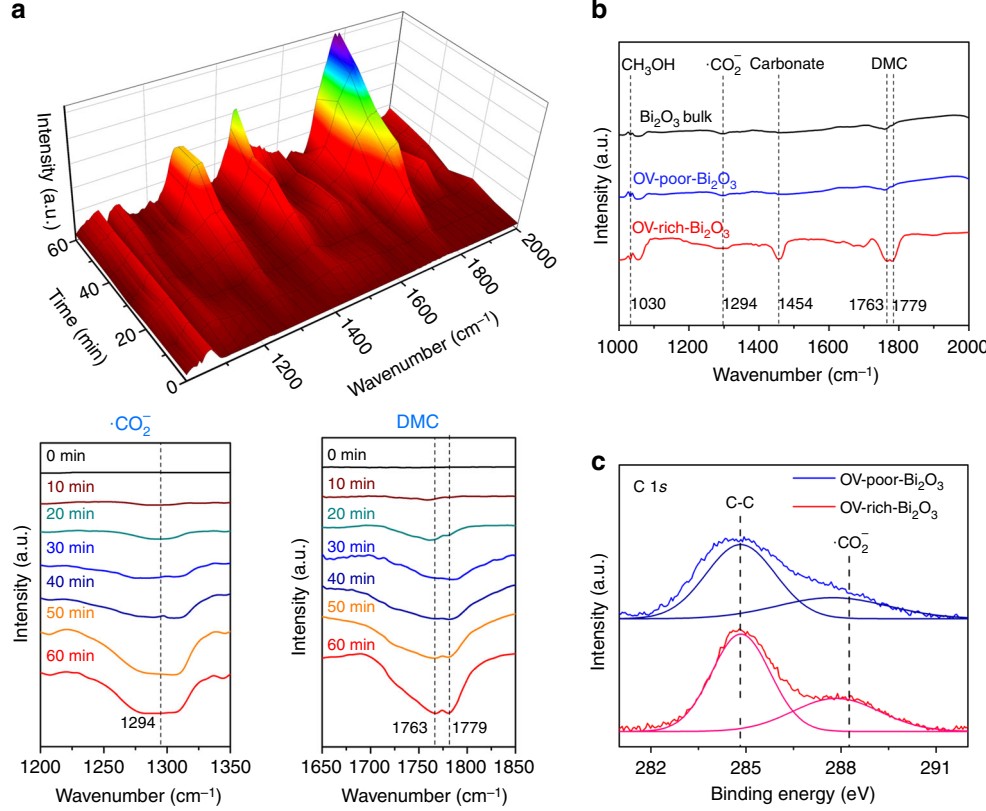

**Fig. 4** Reaction mechanism for $Bi_2O_3$ nanosheets. **a** In situ DRIFTS spectra for the adsorption and activation of $CO_2$ in the presence of $CH_3OH$ under Xe-lamp irradiation for OV-rich-$Bi_2O_3$ nanosheets. Inset at the lower left: the DRIFTS of generation of $\cdot CO_2^-$; inset at the lower right: the DRIFTS of DMC. **b** In situ DRIFTS spectra for the catalysts with $CO_2$ and $CH_3OH$ under Xe-lamp irradiation for 60 min. **c** Quasi in situ XPS spectra of OV-rich-$Bi_2O_3$ and OV-poor-$Bi_2O_3$ nanosheets under the atmosphere of $CO_2$ (0.2 MPa) at 373 K

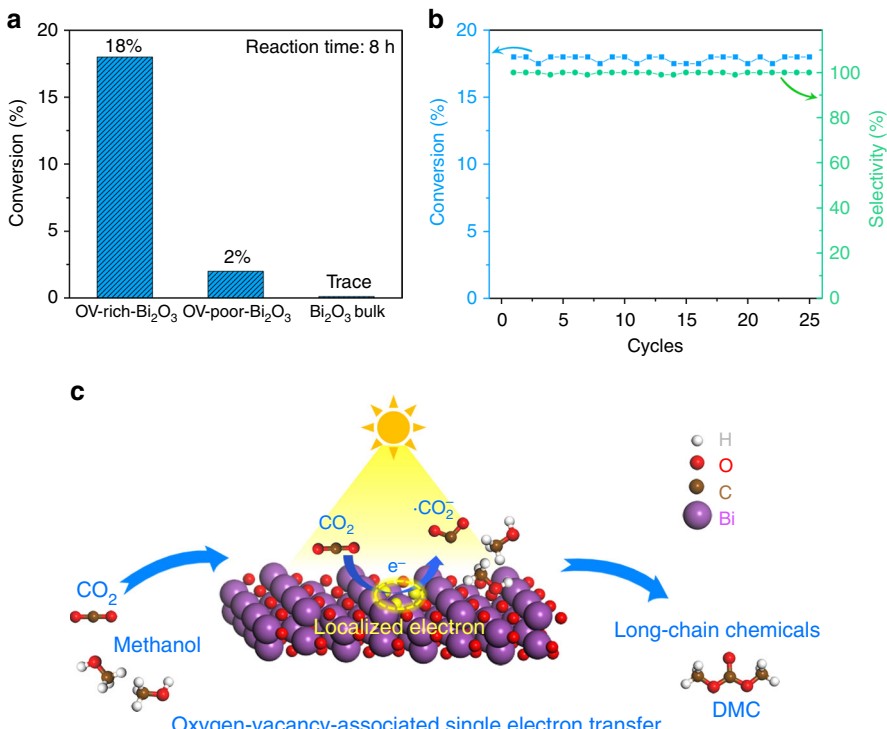

**Fig. 5** Catalytic performances and schematic reaction processes. **a** Performances of various catalysts for $CO_2$ fixation at 373 K under Xe-lamp irradiation. **b** Photostability cyclic test for OV-rich-$Bi_2O_3$ nanosheets. Reaction time for each run: 8 h. **c** Schematic introduction for the photofixation of $CO_2$ to long-chain chemicals

As one of the key factors for photocatalysis, the band structures of the synthetic $Bi_2O_3$ nanosheets were tested to explore the role of OVs in achieving high activity in photoreduction of $CO_2$. Based on the analysis of Mott-Schottky curves, UV–vis spectra and synchrotron radiation photoelectron spectroscopy (SRPES), the conduction band minimum (CBMs) of OV-rich-$Bi_2O_3$ nanosheets and OV-poor-$Bi_2O_3$ nanosheets were determined to be $-0.11$ eV and $-0.08$ eV (vs. normal hydrogen electrode (NHE)), respectively (Supplementary Fig. 10). Interestingly, although the conduction band position of the synthetic $Bi_2O_3$ nanosheets is well below the reduction level for carbonate radical ($CO_2 + e^- \rightarrow \bullet CO_2^-$, $E^\theta = -1.9$ V vs. NHE), the nanosheets could still break the thermodynamic limitations of $CO_2$ activation, further confirming the crucial role of OVs in activating of $CO_2$ on the surface of $Bi_2O_3$ nanosheets.

## Discussion

Based on the combination of theoretical and experimental studies, the reaction for the photofixation of $CO_2$ with methanol to DMC involves four consequently elementary steps described as followings, among which the generation of $\bullet CO_2^-$ group is the rate-limiting step[9,20,31].

$$CH_3OH + h^+ \leftrightarrow \cdot CH_2OH + H^+ \quad (1)$$

$$CH_3OH + e^- \leftrightarrow \cdot CH_3 + OH^- \quad (2)$$

$$CO_2 + e^- \leftrightarrow \cdot CO_2^- \quad (3)$$

$$\cdot CO_2^- + \cdot CH_3 + \cdot CH_2OH \leftrightarrow (CH_3O)_2CO(DMC) + e^- \quad (4)$$

Summary:

$$CO_2 + 2CH_3OH \leftrightarrow (CH_3O)_2CO + H_2O \quad (5)$$

As schematically illustrated in Fig. 5c, the OVs confined in $Bi_2O_3$ nanosheets serve as the efficiently active sites for the adsorption and activation of $CO_2$ to $\bullet CO_2^-$. In detail, the OVs in the surface of $Bi_2O_3$ nanosheets offer abundant coordinatively unsaturated sites for the adsorption of $CO_2$ molecules[25,31]. Meanwhile, the electrons that localized on the OVs are easily to be excited for $CO_2$ activation. In addition, rich OVs could result in the enhanced separation of photogenerated electron and holes in the samples to offer abundant free charge carriers, as illustrated in the photocurrent-voltage curves (Supplementary Fig. 11). Furthermore, $CH_3OH$ molecules intended to be adsorbed on the nanosheets with OVs, exhibiting more negative adsorption energy compared to its vacancy-free counterpart[41,42]. Benefitting from the above advantages, the efficient reduction of $CO_2$ to long-chain chemicals at the OVs in the $Bi_2O_3$ nanosheets could be realized.

In conclusion, we have shown that $Bi_2O_3$ nanosheets with rich OVs are efficient catalysts for the photofixation of $CO_2$ to valuable long-chain chemicals. Theoretical simulations showed that the OVs can not only provide abundant localized electrons, but also lower the adsorption energies of $CO_2$ on the $Bi_2O_3$ atomic layers. Both in situ DRIFT and quasi in situ XPS reveal that OVs in the $Bi_2O_3$ nanosheets could enhance the generation of $\bullet CO_2^-$ species, which is the rate-determining step for $CO_2$ photofixation. As a result, the OV-rich-$Bi_2O_3$ nanosheets could catalyze $CO_2$ and $CH_3OH$ to DMC with extremely high selectivity, whereas only trace amounts of DMC can be detected for pristine bulk $Bi_2O_3$. This work not only paves the way to the design of efficient catalysts for the synthesis of long-chain chemicals, but also

provides insights into the role of defective structures in the $CO_2$ photofixation process.

## Methods

**Materials**. Bismuth powders were purchased from Aladdin. Citric acid (CA, >99.5%) and 5,5-dimethyl-1-pyrroline N-oxide (DMPO, for ESR tests) were purchased from Sigma-Aldrich. D(+)-Glucose (AR), hydrochloric acid (36~38%, AR), anhydrous ethanol (EtOH, AR), ethyl acetate (EtOAc, AR) and acetonitrile ($CH_3CN$, AR) were obtained from Sinopharm Chemical Reagent Co., Ltd. The water used in all experiments was de-ionized (DI). Other chemicals were of analytical grade purity, obtained from Sinopharm Chemical Reagent Co., Ltd. All of the chemical reagents of analytical grade were used as received without further purification.

**Preparation of Oxygen-vacancy-controlled $Bi_2O_3$ nanosheets**. In a typical synthesis, 800 mg Bismuth (Alfa Aesar) was added into a mixed solution of benzyl alcohol (30 mL) and propylamine (6 mL). After vigorous stirring for 20 min, the mixture was then transferred into a 50 ml Teflon-lined autocalve, sealed and heated at 453 K for 12 h. The system was then allowed to cool down to room temperature naturally; the precipitates were collected by centrifugation, washed with ethanol and water for many times then dried in vacuum overnight for further use. Then, 400 mg the following the precipitates was dispersed into 100 ml mixed solution of isopropanol and water (1:1) that bubbled with oxygen to induce the oxidation of bismuth, and the mixture solution was then sonicated at the power of 300 W. The oxygen vacancy-rich ultrathin $Bi_2O_3$ nanosheets (OV-rich-$Bi_2O_3$) were formed by ultrasounding for 1 h. The oxygen vacancy-poor ultrathin $Bi_2O_3$ nanosheets (OV-poor-$Bi_2O_3$) were obtained by ultrasounding for 6 h.

**Preparation of bulk $Bi_2O_3$**. In a typical synthesis, 2 mmol $Na_2SO_4$ and 4 mmol Bi $(NO_3)_3 \cdot 5H_2O$ was dissolved in 30 ml deionized water. Then the pH values of the mixture were adjusted to 11 by adding of 1 M NaOH solution. The mixture was then transferred into a 50 ml Teflon-lined autocalve, sealed and heated at 473 K for 12 h. The system was then allowed to cool down to room temperature naturally; the precipitates were collected by centrifugation, washed with ethanol and water for several times, and then dried in vacuum overnight for further use.

**Characterizations**. XRD spectra was collected by Philips X'Pert Pro Super diffractometer with Cu-Kα radiation ($\lambda = 1.54178$ Å). The HAADF-STEM measurement was carried out on a JEOL JEM-ARF200F. The ESR measurements were recorded on a JES-FA200 model spectrometer operating at the X-band frequency. AFM was measured on the Veeco DI Nano-scope MultiMode V system. The X-ray photoelectron spectra (XPS) were detected on an ESCALAB MKII with Mg Kα as the excitation source, using C 1s (284.6 eV) as a reference. The Fourier-transform infrared (FT-IR) spectra were collected on a MAGNA-IR 750 (Nicolet Instrument Co, U.S.). Room temperature PL spectra were carried out by using a Jobin Yvon Fluorolog 32TAU luminescence spectrometer (Jobin Yvon Instruments Co., Ltd., France). The valence band XPS spectra were detected at beamline BL10B in the National Synchrotron Radiation Laboratory (NSRL), Hefei, China. The NMR experiments were carried out on with a 400-MHz Bruker AVANCE AV III NMR spectrometer. The electrochemical measurements were performed an electrochemical workstation (CHI760E, Shanghai Chenhua Limited, China). Isotope Tracing Experiments: The isotope tracing experiments were performed using $^{13}CO_2$ (99%, Cambridge Isotope Laboratories, Inc.) and the corresponding products were measured by determined via the $^{13}C$-coupled satellites in a 400-MHz Bruker AVANCE AV III NMR spectrometer.

**Calculation method**. The first-principles DFT calculations were carried out with the projected augmented wave method with the Perdew-Burke-Ernzerhof (PBE) GGA functional encoded in the Vienna ab initio simulation package[43]. The convergence on the choice of energy cut-off was tested to 400 eV, and the atomic positions were allowed to relax until the energy and force are $<10^{-4}$ eV and $-0.02$ eV Å$^{-1}$, respectively.

**Catalytic tests**. The direct photogeneration of DMC was performed from the reaction of $CO_2$ and $CH_3OH$ in a 100 mL stainless-steel autoclave. After the addition of 30 mL of acetonitrile ($CH_3CN$) solution and 20 mg catalysts into a Teflon inlet, the autoclave was pressurized with high-purity $CO_2$ pressure (99.99%, 0.2 MPa). The reaction was performed at 373 K with stirring at 400 r.p.m. for 8 h. The light source for the photocatalysis was a 300 W Xe lamp (PLS-SXE300/300UV, Beijing Perfectlight Technology Co., Ltd). The liquid products were detected by NMR (400-MHz Bruker AVANCE AV III) spectroscopy using 1,4-dicyanobenzene as the internal standard.

**In situ DRIFTS measurements**. In situ diffuse reflectance infrared Fourier-transform infrared spectroscopy (DRIFTS) measurements were obtained by using a Bruker IFS 66v Fourier-transform spectrometer equipped with a Harrick diffuse reflectance accessory at the Infrared Spectroscopy and Microspectroscopy Endstation (BL01B) in

NSRL in Hefei, China. The samples were held in a custom-fabricated IR reaction chamber which was specifically designed to examine highly scattering powder samples in the diffuse reflection mode. The chamber was sealed with two ZnSe windows. During the in situ characterization, 0.2 MPa of $CO_2$ was introduced into the chamber and the peaks of free molecular $CO_2$ were set as a reference between the samples. Each spectrum was recorded by averaging 256 scans at a 4 cm$^{-1}$ spectral resolution.

**Quasi in situ XPS measurements.** The quasi in situ XPS measurements were performed at the photoemission end-station at beamline BL10B in the NSRL in Hefei, China. Briefly, the beamline is connected to a bending magnet and covers photon energies from 100 to 1000 eV with a resolving power ($E/\Delta E$) better than 1000. The end-station is composed of four chambers, i.e., analysis chamber, preparation chamber, quick sample load-lock chamber and high pressure reactor. The analysis chamber, with a base pressure of $<5 \times 10^{-10}$ torr, is connected to the beamline and equipped with a VG Scienta R3000 electron energy analyzer and a twin anode X-ray source. After the sample treatment, the reactor can be pumped down to high vacuum ($<10^{-8}$ torr) for sample transfer. In the current work, the samples were treated with the $CO_2$ (0.2 Mpa) at 373 K for 2 h under Xe-lamp irradiation, after which it was transferred to analysis chamber for XPS measurement without exposing to air.

## Data availability

The data that support the findings of this study are available on request from the corresponding authors.

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

## Acknowledgements

This work was supported by the National Key R & D Program on Nano Science & Technology (2017YFA0207301), National Natural Science Foundation of China (U1632149, U1532265, 11621063, 21890754), Anhui Provincial Natural Science

Foundation (1708085QB24, 1808085QB34), the Youth Innovation Promotion Association of CAS (2017493), Young Elite Scientist Sponsorship Program by CAST, and Key Research Program of Frontier Sciences (QYZDY-SSW-SLH011). The authors thank Dr. Yang Xu at Ilmenau University of Technology for useful discussions.

## Author contributions

X.D.Z. and Y.X. supervised the project. X.D.Z., S.C., and H.W. conceived the idea and wrote the paper. S.C., H.W. and S.J. carried the experiments and analyzed the results. Z. K. and B.P. performed the DFT calculations. X.S.Z., Z.Q. and J.Z. collected the synchrotron radiation based spectroscopy data. All authors discussed the results and assisted during manuscript preparation.

## Additional information

**Competing interests:** The authors declare no competing interests.

