## [Peer Review File · Nature Communications]

Reviewers' comments:

Reviewer #1 (Remarks to the Author):

The presented manuscript by Chen and co-authors examines photofixation of CO₂ to long-chain chemicals as shown on the example of DMC synthesis over Bi₂O₃ nanosheets. Although the study represents a clear interest to scientific community, I think the obtained results and their presentation do not warrant publication in Nature Communications in the present form. Below is the summary of my major concerns.

1) English is poor with a lot of grammatical errors/typos. Just to name a few:

Line 38: "CO₂ was often undergoes"

Line 45: "To data"

Line 51: "semiconductor that capable of"

Line 55: "hence active them"

2) The major concern is how stable is the catalyst toward this reaction? The authors only present the conversion rate for 8 cycles, which is very few.

3) As for the conversion rate, no comparison to other catalysts (e.g., CeO_{2-x}) that can convert CO₂ to DMC is provided. This would be very useful and place the performance of Bi₂O₃ into the context of state-of-the-art.

4) It is unclear why the authors state that the generation of CO₂- groups is the rate-limiting step, the references should be provided. Also, for reactions 1) and 2) it should be discussed how oxygen vacancies may affect them.

Reviewer #2 (Remarks to the Author):

In this manuscript, the authors targeted at developing effective catalysts for photofixation of CO₂ to long-chain chemicals. Specifically, by taking Bi₂O₃ nanosheet as a model system, they found that oxygen vacancies confined in atomic layers can lower adsorption energies of CO₂, and thus activate CO₂ by single-electron transfer in mild conditions. In general, this is a well-conceived and carefully performed study on an important question, the result is interesting, and potentially can be extended to other related systems. Publication is recommended after the following issues are considered.

(1) Figure 3 presents the image of the defective Bi₂O₃ nanosheets. Preferably, images confirming the oxygen vacancies and defect structural features can be provided. Such info will provide the basis for future more detailed studies in the community.

(2) ¹³C CO₂ labelling experiment is recommended to confirm that the product is truly from the CO₂ photofixation.

(3) Some typos, eg. Page 1, Line 21, "thus active CO₂ by" -> "thus activate CO₂ by".

Reply Note

Reviewer #1 (Remarks to the Author):

The presented manuscript by Chen and co-authors examines photofixation of CO₂ to long-chain chemicals as shown on the example of DMC synthesis over Bi₂O₃ nanosheets. Although the study represents a clear interest to scientific community, I think the obtained results and their presentation do not warrant publication in Nature Communications in the present form. Below is the summary of my major concerns.

Response: We much appreciate the referee's positive evaluation on the scientific content of our manuscript.

***Comment 1:** English is poor with a lot of grammatical errors/typos. Just to name a few:*

Line 38: "CO₂ was often undergoes"

Line 45: "To data"

Line 51: "semiconductor that capable of"

Line 55: "hence active them"

Response: Many thanks to the referee's kind reminder. We apologize for our careless mistakes. To better present our results, we have asked a native English speaker who has been teaching the English course in our university to have a thorough check and polish the writing of the manuscript. We believe that the **revised manuscript** is now more qualified for publication.

***Comment 2:** The major concern is how stable is the catalyst toward this reaction? The authors only present the conversion rate for 8 cycles, which is very few.*

Response: We gratefully appreciate the referee's kind concern. Stability is a key factor in evaluating the performance of a catalyst. We have further carried out the long-period stability tests for the as-designed catalyst, where the OV-R-Bi₂O₃ nanosheets were tested up to 200 hours. As seen in **Figure N1** below, the high conversion yield and nearly 100% selectivity for DMC generation were kept during continuous 25 cycles, indicating the high stability of OV-R-Bi₂O₃ nanosheets. Furthermore, the crystal phase, morphology and vacancy concentration of the OV-R-Bi₂O₃ nanosheets were well maintained after the long term reaction (**Figure N2**).

Figure N1. Photostability cyclic test for OV-Rich-Bi₂O₃ nanosheets. Reaction time for each cycle: 8 h.

Figure N2. Characterization of the OV-R-Bi₂O₃ sample after catalytic cycles. (a) XRD pattern, (b) Room temperature ESR spectra, (c, d) TEM image of OV-R-Bi₂O₃ before and after catalytic cycles up to 200 hours.

We appreciate the referee's kind suggestions again, and we have modified the stability part of our catalysts in the **revised Manuscript and revised Supporting Information**.

Comment 3: As for the conversion rate, no comparison to other catalysts (e.g., CeO_{2-x}) that can convert CO_2 to DMC is provided. This would be very useful and place the performance of Bi_2O_3 into the context of state-of-the-art.

Response: Many thanks to the referee's thoughtful suggestion. In order to clearly show the catalytic performance of our samples, we have compared them with other well-established catalysts, including CeO_2 , V_2O_5 and ZrO_2 . According to literature, we prepared CeO_2 nanosheets (*Nature Commun.* **4**, 2899, (2013)), V_2O_5 nanosheets (*Adv. Funct. Mater.* **26**, 784–791, (2016)) and ZrO_2 particles (*J. Mol. Liq.* **216**, 342-346, (2016)) (**Figure N3**), and tested their catalytic performances under the same condition to OV-Rich- Bi_2O_3 nanosheets.

Figure N3. Characterizations for other prepared catalysts for comparison. SEM image and TEM image for (a, b) CeO_2 nanosheets; (c, d) V_2O_5 nanosheets; (e, f) ZrO_2 nanoparticles. (g) Performances of OV-Rich- Bi_2O_3 nanosheets, CeO_2 nanosheets, V_2O_5 nanosheets and ZrO_2 nanoparticle for CO_2 fixation under the same reaction conditions (CO_2 (0.2 Mpa) at 373 K under Xe-lamp irradiation).

As seen in **Figure N3g**, the conversion yield of OV-Rich- Bi_2O_3 nanosheets was as high as 18 %, which is much higher than those of the other three catalysts, indicating the outstanding catalytic performance of OV-Rich- Bi_2O_3 nanosheets for DMC generation. In addition, a variety of catalysts that have been reported in previous studies for converting CO_2 to DMC under high pressure and/or high temperature were also compared. As listed in **Table N1** below, it is impressive that the OV-Rich- Bi_2O_3

nanosheets show highest conversion yield and selectivity among the catalysts even under mild conditions without external high pressure and temperature. The above results clearly indicate that the OV-Rich-Bi₂O₃ nanosheets are the state-of-the-art light-driven catalysts for DMC generation.

Table N1. Comparison of performances of various catalysts for direct synthesis of DMC from CO₂ and CH₃OH.

Catalyst	Conversion yield ^a (%)	Selectivity ^b (%)	Reaction condition	Ref.
OV-R-Bi₂O₃ nanosheet	18	>99	0.2 Mpa, 373K, light, 8h	This work
ZrO ₂ -KCl-K ₂ CO ₃	10.8	43.2	9.5 Mpa, 423K, 8h	1
H ₃ PO ₄ /V ₂ O ₅	1.96	93.1	0.6 Mpa, 413K, 8h	2
H ₃ PW ₁₂ O ₄₀ /ZrO ₂	2.02	>99	7.6 Mpa, 373K, 7h	3
CeO ₂ - ZrO ₂	0.8	>99	6 Mpa, 383K, 24h	4
Co _{1.5} PW ₁₂ O ₄₀	3.8	65.2	0.5 Mpa, 373K, 5h	5
Mo-Cu-Fe/SiO ₂	7	88	0.5 Mpa, 393K, 4h	6

(¹ *Ind. Eng. Chem. Res.* **49**, 9609-9617 (2010); ² *J. Mol. Catal. A: Chem.* **238**, 158-162, (2005); ³ *Appl. Catal. A: Gen.* **256**, 203-212 (2003); ⁴ *Appl. Catal. A: Gen.* **237**, 103-109 (2002); ⁵ *Transition Met. Chem.* **35**, 927-931 (2010); ⁶ *Chin. Chem. Lett.* **24**, 307-310 (2013).)

According to the referee's kind suggestions, we have added **Figure N3g** and corresponding discussions to the **revised Supporting Information**.

Comment 4: It is unclear why the authors state that the generation of CO₂⁻ groups is the rate-limiting step, the references should be provided. Also, for reactions 1) and 2)

it should be discussed how oxygen vacancies may affect them.

Response: Many thanks to the referee's kind concern and thoughtful suggestions. The photocatalytic generation of DMC refers to the reaction among $\bullet\text{CO}_2^-$ species, methoxy anion (CH_3O^-) and methylic species (CH_3^+). According to literatures, the activation of CH_3OH by photogenerated electrons and holes to the methylic species and methoxy anion are thermodynamically favorable, because of the high reactivity of CH_3OH molecules (*J. Phys. Chem. C* **119**, 9798-9804, (2015); *J. Catal.* **294**, 199-206, (2012)). However, as for the CO_2 , one of the most thermodynamically stable and kinetically inert molecules (CO_2 with the standard formation enthalpy of $-393.5 \text{ kJ mol}^{-1}$), activating the molecule is recognized to be difficulty due to a high reaction energy (*J. Chem. Phys.* **106**, 1063-1079, (1997); *ChemPhysChem* **18**, 3135-3141, (2017)). Thus, the activation of CO_2 for the generation of $\bullet\text{CO}_2^-$ groups would be the most energy consuming and difficulty step among DMC production. Meanwhile, some literatures suggest that the rate-limiting step in CO_2 photoreduction is the formation of CO_2^- anion by transferring an electron from catalysts to the lowest unoccupied molecular orbital of CO_2 (*J. Phys. Chem. C* **116**, 7904-7912 (2012); *J. Am. Chem. Soc.* **137**, 6393-6399 (2015); *J. Am. Chem. Soc.* **133**, 10066-10069 (2011)).

As for reaction 1) and 2) that referring to the photocatalytic activation of CH_3OH by electrons and holes, respectively, the presence of oxygen vacancies would result in the enhanced separation of photogenerated electrons and holes to offer abundant free charge carriers, as illustrated in the photocurrent-voltage curves (**Supplementary Fig. 11**). In addition, CH_3OH molecules are intended to adsorb on the Bi_2O_3 nanosheets with oxygen vacancies, exhibiting more negative adsorption energy compared to the corresponding vacancy-free sample (*Appl. Surf. Sci.* **253**, 974-982, (2006); *J. Phys. Chem. C* **111**, 10023-10028, (2007)). Thus, the oxygen vacancies in the Bi_2O_3 nanosheets could increase the photocatalytic generation of CH_3O^- and CH_3^+ radicals, which further enhances the efficiency of the reaction.

We much thank the referee for bringing these issues to our attention. Accordingly, we have added the references and corresponding discussions to our **revised Manuscript** and **revised Supporting Information** to clearly illustrate this point.

Once again, we are much appreciated of the referee's kind comments and thoughtful suggestions that greatly improve the quality of our study.

Reviewer #2 (Remarks to the Author):

In this manuscript, the authors targeted at developing effective catalysts for photofixation of CO₂ to long-chain chemicals. Specifically, by taking Bi₂O₃ nanosheet as a model system, they found that oxygen vacancies confined in atomic layers can lower adsorption energies of CO₂, and thus activate CO₂ by single-electron transfer in mild conditions. In general, this is a well-conceived and carefully performed study on an important question, the result is interesting, and potentially can be extended to other related systems. Publication is recommended after the following issues are considered.

Response: We much appreciate the referee's positive evaluation on our study.

***Comment 1:** Figure 3 presents the image of the defective B₂O₃ nanosheets. Preferably, images confirming the oxygen vacancies and defect structural features can be provided. Such info will provide the basis for future more detailed studies in the community.*

Response: Many thanks to the referee's kind suggestion. In our study, the presence of oxygen vacancies in B₂O₃ nanosheets was revealed by the combination of HAADF-STEM image, XPS spectra, ESR spectra and PL spectra, where all the evidences are self-consistent. In detail, the HAADF-STEM image for the defective sample shows obvious lattice disorder (**Figure N4a**). The XPS spectra for O 2p in **Figure N4b** that locates at 530.9 eV could be assigned to the signal of oxygen atoms in the vicinity of an oxygen vacancy (*Nat. Mater.* **10**, 45-50, (2011); *J. Am. Chem. Soc.* **138**, 8928-8935 (2016)). ESR spectroscopy, an effective technique for detecting oxygen vacancies (*J. Am. Chem. Soc.* **138**, 8928-8935 (2016)), was carried out to examine the unpaired electrons that formed by vacancies. As seen in **Figure N4c**, the samples exhibited an obvious ESR signal ($g = 2.002$), which could be identified as the electrons trapped on oxygen vacancies. In addition, beyond the signal of recombination of photogenerated electron-hole pairs (443 nm), an obvious peak for the emission of oxygen vacancies can be detected in the PL spectra (**Figure N4d**).

Despite that it is not feasible to direct visualize the low amount of defects in our sample, the changes in both crystal structure and electronic structure strongly confirms the presence of oxygen vacancies in the as-designed B₂O₃ nanosheets. According to the referee's kind suggestion, we have added more contents about the

features of oxygen vacancies and defect structural in our **revised manuscript**.

Figure N4. Structure characterization for the defect-controlled Bi_2O_3 nanosheets. (a) Atomic-resolution HAADF-STEM images of OV-Rich- Bi_2O_3 nanosheets. (b) O 2p XPS spectra, (c) Room-temperature ESR spectra, and (d) PL spectra of Bi_2O_3 nanosheets with rich and poor oxygen vacancies, respectively.

Comment 2: $^{13}\text{CO}_2$ labelling experiment is recommended to confirm that the product is truly from the CO_2 photofixation.

Response: We gratefully appreciate the referee's thoughtful suggestion. The $^{13}\text{CO}_2$ labeling experiment is a useful tool to reveal the dominant product indeed originated from the photofixation of CO_2 or not. In our original manuscript, part of $^{13}\text{CO}_2$ labeling experiments have carried out for the DMC production (**Supplementary Figure 6**), and we apologize for the unclear illustration on this issue. Herein, we have further carried out the $^{13}\text{CO}_2$ labelling experiment to reexamine previous results. The isotope tracing experiments were performed under the same condition as the parameters in our original manuscript using common CO_2 and $^{13}\text{CO}_2$ (^{13}C , 99 %), respectively. As seen in **Figure N5**, the reaction generates obvious ^{13}C NMR peaks at 156.6-ppm and 54.8-ppm signals referring to DMC, which come from carbon sources

of CO_2 and CH_3OH , respectively. The intensity of $^{13}\text{C}\text{O}_2$ that located at 156.6-ppm is much higher than that of common CO_2 , thus clearly indicating that the product DMC is indeed derived from CO_2 .

We much appreciate the referee's kind suggestions again, and have modified part of the texts of $^{13}\text{C}\text{O}_2$ labeling experiment in the **revised manuscript** and **revised Supporting Information**.

Figure N5. Representative NMR spectra of synthesized DMC from the reaction of CO_2 and CH_3OH in acetonitrile (CH_3CN) solution. ^{13}C -NMR spectra of the products using common CO_2 and ^{13}C isotopic labeling CO_2 .

Comment 3: Some typos, eg. Page 1, Line 21, “thus active CO_2 by” -> “thus activate CO_2 by”.

Response: Many thanks to the referee's kind reminder. We apologize for our careless mistakes. We have thoroughly checked and polished the English of our manuscript with the help of a native English speaker who has been teaching the English course in our university. We believe that the revised manuscript is now more qualified for publication.

Once again, we are much appreciated of the referee's kind comments and thoughtful suggestions in improving the quality of our study.

REVIEWERS' COMMENTS:

Reviewer #2 (Remarks to the Author):

The authors have well addressed the concerns in my previous review. Publication is recommended.